# Studies on Loading Salicylic Acid in Xerogel Films of Crosslinked Hyaluronic Acid

**DOI:** 10.3390/gels10010054

**Published:** 2024-01-11

**Authors:** Anastasia Maria Mamaligka, Kalliopi Dodou

**Affiliations:** School of Health and Life Sciences, University of Teesside, Middlesborough TS13BX, UK

**Keywords:** hyaluronic acid, salicylic acid, xerogels, drug loading, hydrogels

## Abstract

During the last decades, salicylic acid (SA) and hyaluronic acid (HA) have been studied for a wide range of cosmetic and pharmaceutical applications. The current study investigated the drug loading potential of SA in HA-based crosslinked hydrogel films using a post-loading (osmosis) method of the unmedicated xerogels from saturated aqueous solutions of salicylic acid over a range of pH values. The films were characterized with Fourier-transform infra-red spectroscopy (FT-IR) and ultraviolet-visible (UV-Vis) spectrophotometry in order to elucidate the drug loading profile and the films’ integrity during the loading process. Additional studies on their weight loss (%), gel fraction (%), thickness increase (%) and swelling (%) were performed. Overall, the studies showed significant film disintegration at highly acidic and basic solutions. No drug loading occurred at neutral and basic pH, possibly due to the anionic repulsion between SA and HA, whereas at, pH 2.1, the drug loading was promising and could be detected via UV-Vis analysis of the medicated solutions, with the SA concentration in the xerogel films at 28% *w*/*w*.

## 1. Introduction

In the last decades, transdermal patches have been a fast-emerging drug delivery system. Their wide use in the cosmetic and pharmaceutical industries is supported by the scientific knowledge acquired over recent decades and the advances in fields such as nanotechnology and material science [1,2]. The latest findings enabled the development of novel patches and the incorporation of numerous ingredients [2,3,4,5]. More studies should be conducted in order to unlock the limitless potentials in this field.

Patches are semisolid systems that belong to the category of polymeric gels [6]. Traditionally, patches consisted of hydrophobic pressure-sensitive adhesive polymers; in recent years, the use of hydrophilic polymers has also been explored. Hydrogels are an effective and environmentally friendly vehicle for the loading and release of active ingredients [7,8,9]. They are polymeric, swellable 3D networks able to absorb large amounts of water or biological fluids [10]. At the same time, they are insoluble due to the presence of covalent crosslinks. Hydrogels consist of a polymer, a liquid and a crosslinking agent that initiates the formation of the 3D network [11]. Crosslinks can be formed by chemical reactions initiated by heat, pressure, change in pH, or irradiation. The hydrogels can be loaded with active ingredients either during their preparation (in situ loading) or after (post-loading or osmosis) [12].

The semi-solid hydrogels can be reversibly converted to xerogels via water evaporation, e.g., using convective drying; during this process, the gel maintains its 3D network structure [13,14]. Xerogels are a fascinating area of research for multiple industries due to their unique and attractive properties. One of their main characteristics is their high porosity and large surface area with small pore size, high density and mechanical stability as well as inexpensive fabrication pathways [15]. They are synthesized from numerous hydrophilic polymers such as chitosan [16], sodium alginate [17], pectin [18], cellulose [19] and hyaluronic acid [12]. They have been explored within the food and pharmaceutical industries for their use as encapsulating agents, delivery vehicles and others [13]. They find wide medical and biomedical applications due to their high biocompatibility, non-immunogenicity and non-cytotoxicity. They are widely used in tissue engineering [20] and as drug delivery systems as they possess high drug loading capacity and their porosity enables the sustained drug release of active ingredients [12,21,22].

The need to protect the environment is becoming increasingly critical. Therefore, all industries are focused on the development of environmentally friendly products. The pharmaceutical and cosmetic industries focus on the replacement of synthetic polymers with biodegradable polymers such as sodium alginate, hyaluronic acid, collagen and others, which can be loaded with various active ingredients and are biodegradable [2]. HA is one of those hydrophilic polymers that has attracted a large commercial interest. Numerous studies have been published on the simultaneous use of HA with salicylic acid (SA) for various biomedical applications such as electrospun membranes comprised from HA and loaded with SA [23], hydrogels with in situ encapsulation of HA for controlled drug release [24], HA and gelatin clay composite hydrogels for cell adhesion and controlled drug delivery [25], cosmeceutical products [26], dissolvable microneedles [27] and others.

Hyaluronic acid (HA) or hyaluronan is an anionic linear polysaccharide which consists of repeating units of (β,1-4)-D-glucuronic acid and (β,1-3)-N-acetyl-D-glucosamine [28] (Figure 1). In its primary form (n-HA), it is a polymer with molecular weight greater than 2 × 10^4^ kDa; it is found in the extracellular matrix of all animal tissues and is also a significant constituent of the skin [29]. HA has a pKa value of about 3 and is a polyanion associated with cations [30]. Thus, at pH 3, it is 50% ionized and, at higher pH, its ionization degree increases. During the last 40 years, numerous studies have emphasized on HA and its potential applications for the treatment of inflammatory conditions, for wound healing and cell growth [28] and for its rejuvenating effects on the skin [31]. It finds application in many fields due to its biodegradability, biocompatibility, lack of toxicity, ease of chemical modification and high potential drug loading as well as due to its commercial availability in various forms such as gels, tubes, powder and others [29,32]. It is a valuable molecule with applications in arthritis, ophthalmology [33], tissue engineering [34], cosmetic formulations [27,35] and dermal filler injections [36,37]. HA is successfully employed in drug delivery applications as conjugates or as hydrogel depot systems [29,38,39]. It is used in patches both to perform its effects as an active ingredient (low-molecular-weight HA) and also as a building block (high-molecular-weight HA) for the hydrogel 3D network [40]. It has, additionally, been extensively studied as the main component of dissolvable microneedles [35].

In aqueous solutions, high-molecular-weight HA presents significant viscoelastic properties. Its high number of polar groups provide to the molecule high water binding capacity [28,40]. Its hydroxyl, carboxylic and amide groups are available to form bonds and enable crosslinking with several crosslinking agents [32]. However, HA exhibits specific chemical properties that hinder its use. It is a sensitive and highly soluble molecule that can easily undergo degradation due to several physical or chemical factors such as heat, pH, oxidation and others [28]. Chemically crosslinking HA is a common practice used in order to enhance its mechanical properties, solubility and degradation profile due to the covalent bridges and intermolecular bonds that are formed between HA and the crosslinking agent [32,40]. Several studies have been conducted on developing physical or chemical crosslinked HA as novel drug delivery systems. In one of our research group’s previous studies, we developed novel crosslinked hydrogel films with Pentaerythritol Tetra-acrylate as a crosslinking agent [32]. The pH-related degradation of HA is proven by numerous studies both in the unprocessed molecule but also in chemically crosslinked HA-based gels [41]. The existing studies reported high degradation of HA at highly acidic and highly basic pH due to acidic or alkaline hydrolysis and protonation of glucuronic groups [28,42,43]. According to the literature, degradation is higher at HA of high molecular weight [28].

One of the most studied actives for drug loading applications is salicylic acid (SA), a beta-hydroxy acid and the precursor molecule of aspirin (acetylsalicylic acid) (Figure 1) [23,25]. It is the most widely used analgesic and anti-inflammatory agent in the world. At the same time, it has a 2000-year history as an ingredient used for the treatment of skin disorders due to its keratolytic, bacteriostatic, fungicidal and photoprotective properties [44,45,46]. However, the main emphasis is given to its keratolytic effect as its topical application reduces the rate of keratinocyte proliferation and solubilizes the SC by dissolving the intercellular cement [45]. One of the most common uses of SA is for acne treatment in topical formulations or patches [46]. Anti-acne patches emphasize on absorbing pimple fluids and removing dirt and sebum that could aggravate surface-level acne [47]. They can enhance the recovery process both in terms of speed and quality and prevent scar formation. SA has also excellent exfoliating properties enabling its use for indications such as melasma, photodamage and others [44]. SA is considered safe for topical application and its safety and efficacy are well documented for all skin types [44].

SA is a weak acid with pH-dependent aqueous solubility. This property makes SA an interesting molecule as it can interact with biological membranes or be present in aqueous solution depending on its ionization status [48]. Salicylic acid has a pka = 2.97, and its ionization and solubility in water increase with pH increase (Table 1) [48].

The aim of the current study was to evaluate the loading potential of SA in crosslinked HA xerogels and explore the film integrity during loading or swelling in SA aqueous solutions within a wide pH range. The methods used included UV-Vis spectrophotometry, FT-IR spectroscopy and complementary weight and film thickness studies. SA was selected as an ingredient that has high aqueous solubility, provides a simple model for transport studies and is widely used in transdermal patches. To the best of our knowledge, the investigation of the drug loading potential of SA in hydrogel films via post-loading method (osmosis) has not been reported before.

## 2. Results and Discussion

### 2.1. Determination of SA Saturated Solubility

The saturated solubility of SA in water varies at different pH values and, more specifically, it increases proportionally with the degree of ionization of SA [48]. It has been reported that the ionization of SA in aqueous solutions increases from 9.9% at pH 2 to 91.5% at pH 4 and 99.1% at pH 5 at 37 °C (Table 1) [48]. Thus, since SA is almost completely ionized at pH 5, at pH 7.8 and 11 it is expected to be 100% ionized and its saturated solubility to be the same. This was verified experimentally. The saturated solubility of SA in water with different pH values at 25 °C was determined with the method analyzed in Section 4.2.2. and, from Equation (2), it was found to be 1.97 mg SA/mL H_2_O at pH 2.1 and 2.47 mg/mL H_2_O at pH 7.8 and 11 (Table 2). As expected, the solubility of SA in aqueous solutions increased with increasing pH and, thus, higher ionization degrees. This can be explained by the fact that the ionized form of a substance is more soluble than the free acid due to the free units available to form bonds with water molecules. SA is an acid with pKa~2.97 [48]. As such, at pH values higher than 2.97, SA loses H^+^ from its carboxy- groups and becomes ionized and negatively charged (Table 1). However, molecules in the same ionization state have the same saturated solubility (pH 7.8 and 11). In this study, the determined saturated solubility was lower for the same pH values compared to the literature [48]. This was attributed to the lower temperature (25 °C) in which the experiment was conducted compared to the study of Otto et al. (37 °C). It has been reported that the solubility of salicylic acid in water increases at higher temperatures [49].

### 2.2. UV-Vis Spectrophotometry

#### 2.2.1. Calculation of λmax

Calculation of λmax is important as it increases sensitivity and ensures minimum deviation from Beer–Lambert’s law. For the determination of the λmax of SA and HA sodium salt, three solutions at different pH were prepared for each substance and samples were analyzed spectrophotometrically. The UV-Vis spectrum graphs for each solution are presented in Figure 2.

In the case of SA solutions, multiple peaks were observed (Figure 2d–f). In all cases, the peak with the higher absorbance was taken into consideration and it was revealed that the λmax (246 nm) was not pH-dependent and was the same in every test solution.

For the HA solutions, the λmax increased proportionally with the pH. At pH 2.1, the sample presented one peak at 208 nm (Figure 2a). At pH 7.8 and 11 (Figure 2b,c), the samples presented two peaks each. The peaks with the higher absorbance values were not the ones selected as they were at low wavelength and close to the measuring limit of the spectrophotometer. The peaks at 256 nm and 296 nm were the ones selected for pH 7.8 and 11, respectively. One of the parameters that affect UV-Vis absorbance is the pH of the solution as it can affect the chemical structure of the chromophore groups, which may present their maximum peaks at a different wavelength [50]. At low pH, the sodium carboxylate salt of the HA shifted equilibrium towards the free protonated carboxylic acid, whereas, at pH 7.8 and 11, there was only the unprotonated sodium carboxylate form present. This is evidenced by the spectra in Figure 2; the intensity and value of the λmax at pH 2.1 (Figure 2a) is different from the λmax at pH 7.8 and 11 (Figure 2b,c). We should add that the contribution of the hydroxyl group of HA on these spectra is not affected by the these pH values considering that its pKa is about 13; therefore, it is present at its protonated, unionized form at all these studied pH values [51].

#### 2.2.2. Determination of Drug Loading

The xerogels were loaded using the post-loading method described in Section 4.2.1. Xerogels were immersed in saturated SA solutions with concentrations of 1.97 mg/mL for pH 2.1 and 2.47 mg/mL for pH 7.8 and 11. After the withdrawal of films, the loading solutions were scanned and their absorbance values are presented in Figure 3. Statistical analysis of the absorbance data presented small standard deviation between the three measurements for each solution (n = 3). If drug loading occurred, the post-loading concentration would be lower compared to the pre-loading one due to the SA molecules migrating from the solution and entering the film 3D network. According to Beer–Lambert’s Law, the same pattern is expected for the absorbances of these solutions [12].

A paired *t*-test was performed to evaluate whether there is statistically significant difference between the absorbance values before and after loading. The *p*-value referring to pH 2.1 and 11 was less than 0.05 and the difference between the sample groups was considered statistically significant. However, for pH 7.8 and 11, the mean absorbance post-loading was higher than the mean pre-loading. This could be due to instrumental error as the concentration and absorbance post-loading is expected to be the same or less compared to pre-loading. These results demonstrated that the films did not present SA solution uptake and drug loading did not occur at pH 7.8 and 11. However, at pH 2.1 the absorbance values post-loading were lower than before and their difference was statistically significant (*p* < 0.05). This could indicate that limited loading did happen.

For further investigation of the loading at pH 2.1, the calibration curve of SA at pH 2.1 was plotted. The UV-Vis spectrophotometer was adjusted to the correct λmax (246 nm) and scans of the standard solutions were taken. The generated calibration curve for SA is presented in Figure 4. The concentration of the SA solutions pre- and post-loading were calculated based on the linear regression equation and are presented in Figure 5. The lower post-loading concentration is indicative of loading of SA in the films. The mass fraction of SA in the xerogel was found to be 0.28 (28% *w*/*w*).

In addition, the ionization (%) of HA was investigated for the better understanding of the drug loading results. HA and SA are weak acids with functional groups that are easily ionized in pH > pKa. The ionization degree of HA in water with different pH was calculated from Equation (4) and its values are presented in Table 3. The ionization degrees of SA in different pH are presented in Table 1 [48]. It is evident that both HA and SA are fully ionized at pH 7.8 and 11 and the unsuccessful loading could be a result of the anionic repulsion between the two compounds [52]. At pH 2.1, SA is 11.18% ionized and HA around 9.9%. The lower ionization of the compounds in pH < pKa could have permitted the drug loading of SA [52]. However, this was not observable by the FT-IR data.

### 2.3. Fourier-Transform Infra-Red (FT-IR)

The FT-IR method is widely used to identify changes in the characteristic peak position of a compound, structural features and interactions of different groups in a molecule. In this study, it was used to investigate the potential loading of SA within the studied films.

The FT-IR spectra of films loaded in SA solutions or water (Blank) with different pH values are presented in pairs in Figure 6. Figure 7 displays the spectra of pure SA powder. It is observed that, for each pH, the spectra for the loaded and blank films are almost identical. In addition, no noticeable differences are observed between the aforementioned spectra and the spectra presented in the study of Rashid et al. that refer to the same xerogels pre-loading [32]. The peak 1750–1350 cm^−1^ observed at pH 7.8 and 11 is not observed at pH 2.1. In that case, it is replaced by multiple peaks that resemble the spectra of SA powder (Figure 7). However, these data do not prove that drug loading occurred as the same peaks are observed at the spectra of the blank films. Thus, this change is associated solely with the effect of highly acidic pH on the films. These spectra are presented collectively in Figure 8. The peak at 1600^−1^–1680 cm^−1^ is indicative of the C=C group stretch [53,54]. The peak at 1738 cm^−1^ can be attributed to the stretching vibration of C=O groups of Pentaerythritol Tetra-acrylate, which is the cross-linking agent of the films [32]. At 1100 cm^−1^, the characteristic ether C-O bond stretching is noticed as part of the methyl vinyl ether unit (C-O-C) that was created during the reaction between carbonyl carbon of the cross-linker and the -OH group of HA [32,55]. The broad band near 3300 cm^−1^ belongs to the OH^−^ groups [56].

Based on the FT-IR data, drug loading did not occur at any pH as the presence of SA in the films would result in noticeable differences between the spectra of the blank samples and the ones immersed in SA solutions, but the absence of such peaks could be attributed to the low concentration of SA in the films alongside the low sensitivity of this method.

### 2.4. Weight Studies

Weight studies were necessary in order to evaluate the potential drug loading of SA and the films’ integrity within the solutions. Table 4 presents the weight of xerogels before loading, swelled films after loading, swelled films after drying in the oven for 24 h (xerogel post-loading A) and the same xerogels after removal from the oven and conservation for 5 days in room environment conditions (xerogel post-loading B). The xerogels used in this study had high weight uniformity with an average initial weight of 0.0058 g (±0.00113 SD, n = 22). The scale used had a resolution of 2–10 mg and uncertainty of ±0.1 mg [57].

Τhe films were loaded in 10 mL of saturated SA solutions with concentrations of 1.97 mg/mL for pH 2.1 and 2.47 mg/mL for pH 7.8 and 11. Thus, if 100% drug loading occurred, the maximum xerogel weight increase would be 0.0197 g at pH 2.1 and 0.0247 g at pH 7.8 and 11. In all cases, the xerogel weight decreased post-loading. As a result, calculation of drug loading (%) from Equation 3 generated negative values. This could be indicative of the film’s partial disintegration. Therefore, the evaluation of drug loading (%) based on weight measurements of the film cannot provide accurate results.

Weight Loss A (WL_A_) is the film’s weight loss compared to its dry weight right after the oven (xerogel post-loading A), whereas Weight Loss B (WL_B_) is the film’s weight loss compared to its dry weight after removal from the oven and conservation in room environment conditions for 5 days (xerogel post-loading B). Significant film weight loss (WL_A_ or WL_B_) occurred in SA solutions and water at all pH values (Table 5). Films loaded in SA solutions presented the most significant weight loss at pH 11 and pH 2.1. At pH 7.8, weight loss was significantly lower. Blank samples did not present significant weight loss differences among the solutions. The noticed weight loss could be attributed to the partial disintegration of the films in the solutions according to the following literature; according to Maleki et al., in aqueous solutions, HA presents higher disintegration at pH < 4 and pH > 11, while, in the in between values, the HA chains are not affected [43]. Additionally, it has been reported that disintegration is higher for high-molecular-weight HA (around 3000 KDA) [28]. This further explains the high degradation of HA during loading, since the xerogels were prepared with HA sodium salt of high molecular weight (1800–2200 KDa) [32]. The molecular weight of HA is considered high when it is more than 1000 Kda [58]. The degradation at pH 2.1 and 11 can be attributed to acidic or alkaline ether cleavage of the crosslinked HA [28,32,41]. During the crosslinking of the studied films, ether bonds between the hydroxyl group of the HA and the carbonyl oxygen of the Pentaerythritol Tetra-acrylate were formed. The HA chain degradation at highly acidic pH was, additionally, attributed to the protonation and hydrolysis that occurs on the glucuronic acid residue [28]. At highly basic pH, it was attributed to hydrolysis on the N-acetylglucosamine residue. It has been reported that the hydrolysis discussed follows first-order kinetics [59]. At the same time, random HA chain scission could have also affected the film integrity in the test solutions [59].

It can be noticed that WL_B_ (%) is lower than WL_A_ (%) (Figure 9). These results were expected as the xerogels’ weight increased after removal from the oven and conservation in room environment conditions. Due to its high hygroscopicity, HA absorbs moisture from the environment up to 1000 times its solid volume [60]. These data showcase the importance of storing hyaluronic acid films in sealed containers to avoid excessive water uptake during experiments or throughout their commercial shelf life [60]. The crosslinks of the studied xerogels are formed between the hydroxyl group of HA and the carbonyl carbon of Pentaerythritol Tetra-acrylate (ether bonds) [32]. Meanwhile, water molecules from the environment bind with the hydroxyl groups of HA with hydrogen bonds [61]. Higher WL_A_-WL_B_ difference indicates higher moisture uptake from the environment, which could be related to alterations in the films’ ether bonds that enabled more hydroxyl groups of HA to bind with water (Table 6). Ether cleavage can occur under acidic and basic conditions, but the data from this study did not showcase significant pH dependency for films swelled in water as the WL_A_-WL_B_ difference varied [32,41] (Figure 9). However, for films swelled in SA solutions, the WL_A_-WL_B_ difference increased with increasing pH.

Previous studies reported that swelling is affected by the film composition and the loading solution [62]. This was experimentally investigated and verified in the current study. It was found that, in all test solutions, the swelling degree (%) of the films increased with pH increase and the highest swelling was noted for the most ionized states of the films (Figure 10). The films loaded in SA solutions or water presented a similar swelling increase (%) in all cases except pH 11, where swelling increase was significantly higher at the SA-loaded samples. This was in accordance with the higher weight loss for the films and indicated a potential destructive effect of SA on the HA chains and the bonds in the 3D network of the crosslinked films. The pH-dependent swelling values of the films were due to the different ionic strength of each loading solution [48]. HA is a weak polyelectrolyte and, thus, is highly sensitive to pH and ionic strength [63]. These data agree with the weight loss values which were high at pH 11 and 2.1 and proved that the swelling behavior of the studied films is pH-dependent.

Gel fraction A (GF_A_) (%) was calculated from the xerogel weight straight after removal from the oven and gel fraction B (GF_B_) from the xerogel weight after xerogels were conserved in room environment conditions for 5 days (Equation (8)). As expected, GF_B_ is higher than GF_A_ since the xerogel weight post-loading A is lower than the weight post-loading B (Figure 11). The highest gel fraction was noticed for films immersed in solutions with pH 7.8, where the lowest weight loss was observed. This was indicative of lower disintegration compared to the highly basic and acidic solutions [32]. Data from Table 5 showcase that the values of GF (%) and WL (%) are inversely proportional.

### 2.5. Film Thickness

The initial xerogel thickness (X_1_) was the same for all samples as they were cut from the same initial xerogel film. All thickness results are presented in Table 7. It was noticed that each sample had uniform thickness throughout the surface with low standard deviation (≤0.01). In addition, the thickness increase (%) was similar for all samples and did not change significantly between films loaded with different solutions. It was found to be almost double in every case between post- and pre-loading (Figure 12). Between the SA samples, the highest increase was noticed for films immersed in solutions with pH 2.1 and 11, where highest HA film decomposition was reported (Figure 12). This could be attributed to the fact that SA solution entered the polymer matrix and the chains began to relax and to turn into oligomers and monomers as has been reported in the case of other types of hydrogel films [2].

### 2.6. Discussion

During this research, weight studies were successfully employed and contributed in the characterization of films’ integrity within water and aqueous SA solutions with different pH values. Our results demonstrated higher xerogel weight loss for samples immersed in solutions with pH 2.1 and pH 11. This was attributed to the higher disruption of the HA chains due to acidic or alkaline ether cleavage, while the degradation at low pH was additionally linked to the protonation of glucuronic acid groups [28,43]. The weight loss could, additionally, be caused by the scission of HA due to the cleavage of glycosidic (covalent) bonds. These factors could have resulted in the disruption of the films’ 3D network. The correlation between film disintegration and weight loss is confirmed by several studies that proved the higher degree of HA degradation at pH < 4 and pH > 11 [28,43].

The difference between WL_A_ (%) and WL_B_ (%) due to the weight increase of xerogels during storage in room environment conditions showcased the hygroscopicity of the crosslinked xerogels and highlighted the importance of storing these films in containers with controlled humidity throughout a study. The highest gel fraction was noticed for samples loaded in pH 7.8, where the lowest weight loss was observed, in agreement with the hypothesis that disintegration of the films was induced in acidic and basic conditions. All films presented a similar thickness increase after loading, varying between 86 and 113%.

These results are presented collectively in Figure 13 and Figure 14. It can be noticed that the highest swelling (%) is reported for films loaded in SA at pH 11, the thickness increase (%) presents similar values for all samples, the highest weight loss (%) was recorded for films loaded in SA at pH 2.1 and 11 and the highest gel fraction was reported at SA samples immersed in pH 7.8. It is interesting to note that all blank samples (films swollen in water) presented similar values at every pH in comparison to the loaded films indicating that, apart from pH sensitivity, HA films are also sensitive to the loading solution, as an effect of SA on the studied films was demonstrated. Further investigation is required to precisely describe the effect that SA can have on HA.

The determination of SA loading in the studied xerogels with UV-Vis spectrophotometry showcased that the absorbance values pre- and post-loading were similar for pH 7.8 and 11 and proved that drug loading did not occur due to the complete ionization of both compounds, which resulted in anionic repulsion between them. For future studies that aim to perform drug loading with the osmosis method, investigation of the ionization state of the model drug and the film-component polymer is strongly advised. At pH 2.1, the UV absorbance and the concentration of the post-loading SA solution was lower with statistical significance, indicating possible SA intake in the films. However, this was refuted by the FT-IR analysis, which concluded that the spectra of loaded and non-loaded films were identical for all cases. The evaluation of drug loading with weight studies was not possible due to the film weight loss attributed to film disintegration. Although further investigations are needed, the present study does not suggest evaluation of drug loading using weight studies for films composed of compounds with high pH sensitivity (i.e., hyaluronic acid) due to their susceptibility to degradation.

All studies concluded that the highest degradation is noticed for films loaded in SA solutions with pH 2.1 and pH 11. The fact that drug loading did not occur at pH 7.8 and 11 was attributed to the anionic repulsion between SA and HA. However, the loading at pH 2.1 is more promising due to the lower ionization degree of the studied compounds. The high pH dependency for the HA films was proven by all the conducted studies.

Previous studies have reported that the covalent bridges and intermolecular bonds formed during the crosslinking of HA provide to the molecule enhanced mechanical properties and degradation profile [32,40]. However, the current weight studies proved the pH sensitivity of crosslinked HA films and their potential degradation in highly acidic and basic pH values. For future development of HA films, studies related to the behavior of HA films in solutions with different pH values are considered necessary.

Future studies should evaluate the drug loading efficiency of other acidic compounds (i.e., lactic and glycolic acid) in hyaluronic-based films to further elucidate and establish the effect of anionic repulsion during the post-loading (osmosis) of films. Studies could be repeated as a function of time over a longer period of time in order to define the kinetics of degradation for the studied films. The crosslinks present in HA hydrogels prevent HA dissolution in water. However, additional studies can be conducted so as to minimize the film disintegration in extreme pH values and broaden their future application.

## 3. Conclusions

HA and SA are two compounds of great interest for the modern cosmetic and pharmaceutical industries with promising applications. This study focused on the evaluation of the drug loading potential of saturated SA solutions in HA-based films and the films’ integrity within acidic, neutral and basic SA solutions and water. The drug loading efficacy was evaluated using UV-Vis spectrophotometry of the medicated solutions, using FT-IR spectroscopy of the xerogels and with complementary weight and film thickness studies. The results from FT-IR, weight and UV-Vis studies proved that films did not present any SA uptake during loading at pH 7.8 and 11, presumably due to anionic repulsion between SA and HA. At pH 2.1, the drug loading was promising, considering the lesser ionic repulsion forces; this hypothesis was confirmed by the UV-Vis results, whereas the FT-IR and weight studies were inconclusive due to low sensitivity and film degradation respectively. The swelling (%), weight loss (%) and thickness increase (%) values presented the highest values at highly basic and acidic pH, which indicated the higher degradation of HA and disintegration of the HA-based films at those pH regions. The examined films were found to be highly sensitive to pH and ionic strength and all measured parameters were found to be pH-dependent. It was, therefore, concluded that drug-content analysis of the medicated solutions, before and after xerogel immersion, is more accurate than the analysis of the xerogels due to the structural instability of the latter during the drug loading process.

## 4. Materials and Methods

### 4.1. Materials

Salicylic acid crystalline powder (>99.9% salicylic acid) was supplied from BDH Chemicals Ltd. (Poole, UK) and was used for the preparation of salicylic acid solutions. HA sodium salt (sodium hyaluronate; 1800–2200 Kda molecular weight) was sourced from Infinity Ingredients (Binfield, UK). Distilled water was obtained from the water distillation assembly in laboratory 105 of the Faculty of Health Sciences and Wellbeing, University of Sunderland (Sunderland, UK). Sodium hydroxide (18% sodium hydroxide aqueous solution) and citric acid (50% citric acid aqueous solution) were used for pH adjustments. The crosslinked hyaluronic acid xerogels were prepared according to the method described in our previous studies [32]. The films consisted of 5% *w*/*w* HA and 1.25% *w*/*w* PT as a crosslinking agent and their preparation included oven-assisted thermal crosslinking.

### 4.2. Methods

#### 4.2.1. Drug Loading Method

A post-loading (osmosis) method was used to investigate the drug loading potential of SA in the xerogels of interest and the film integrity in the selected solutions. The xerogels were already prepared and precut in 0.5 cm × 0.5 cm rectangular shapes. Each xerogel was added to a glass vial containing 10 mL water (blank samples) or 10 mL SA aqueous saturated solution and left in a water bath at 32 °C for 24 h [64]. Usually, drug loading is conducted at 37 °C. However, due to the thermal sensitivity of HA, a lower loading temperature (32 °C) was chosen [59,65]. HA undergoes irreversible degradation at 1 < pH < 11 [59,65]. Thus, in order to investigate the effect of pH in drug loading, all tests were repeated in solutions with pH 2.1, 7.8 and 11. The value 2.1 was the pH of an SA-saturated solution without pH adjustment, 7.8 was chosen as a mid-range pH that was identical with the pH of the distilled water used and 11 was chosen as a very basic pH. The experiments were repeated for n = 3–5.

#### 4.2.2. Determination of SA Saturated Solubility

The theoretical saturated solubility of SA in water is 0.224 g/100 mL at 25 °C [66]. The true saturated solubility was determined experimentally. For this cause, 0.224 g SA was added in a 100 mL volumetric flask, topped up with distilled water and the solution was bath sonicated for 20 min at room temperature for complete SA dissolution (Transsonic T460, Camlab, Cambridge, UK). The presence of visible insoluble matter (solid SA) indicated the supersaturated state of the solution. For the quantification of the saturated solubility of SA, the solution was filtered twice with pre-weighted filter paper and clarity increased significantly. The resultant solution was passed through a syringe filter with surfactant-free cellulose acetate (SFCA) membrane and 0.45 μm pore size and a clear solution with pH 2.1 was received. The filters were left to dry for 72 h in room environment conditions and their weight was measured. The weight of the SA solids was calculated according to Equation (1):W_SA1_ = W_f2_ − W_f1_(1)
where W_SA1_ is the weight of the solid SA on the filter paper, W_f1_ the weight of the dry filters before filtering and W_f2_ the weight of the dry filters containing the solid SA.

Following this, the saturated solubility of SA was calculated according to Equation (2):W_SS_ = W_SA2_ − W_SA1_(2)
where W_SS_ is the weight of SA needed to create a 100 mL saturated solution, W_SA2_ the initial weight of SA added in water and W_SA1_ the weight of solid SA on the filters. The saturated solubility of SA in water was expected to be the same at pH 7.8 and pH 11 [48]. To verify that an SA-saturated solution with pH 2.1 was prepared, additional amounts of SA (0.01 g each time) were added, pH was adjusted to 7.8 ± 0.01 after each addition with 18% NaOH or 50% citric acid aqueous solutions and bath sonication followed to ensure complete dissolution. The same SA quantity was used to prepare a saturated SA solution with pH 11 after the needed pH adjustments.

All weight measurements during the experiments were conducted in a 40SM-200A Precision Digital Scale Balance 40G/200G (Precisa, Livingston, UK).

#### 4.2.3. UV-Vis Spectrophotometry

UV-Vis spectrophotometry was employed in order to determine the λmax of HA and SA and evaluate the drug loading potential of SA in HA-based films. The absorbance values of the solutions before and after loading were measured at each pH in order to quantify the drug loading.

##### Determination of λmax

The λmax of aqueous SA solutions at pH 2.1, 7.8 and 11 was determined using an AtiUnicam UV2 UV/VIS Spectrometer (Baltimore, MD, USA). For this cause, three SA solutions were prepared (Table 8). For the preparation of each one, 0.01 g SA was added to a 100 mL volumetric flask and was toped up with distilled water. After complete dissolution with magnetic stirring for 30 min, the pH of each solution was adjusted. A sample of each was added in a cuvette (IR Quartz, 45 × 12.5 × 12.5) and their λmax was determined. The UV-Vis Spectrophotometer was operated in absorbance mode, from 190 nm (start lambda) to 300 nm (stop lambda), 2.0 nm bandwidth, 120 nm/min scan speed, 2.0 nm data interval, 325 nm lamp change, for 1 cycle, at a resolution of 0.5 nm.

##### Plotting of Calibration Curves

After determination of the λmax, the UV-Vis spectrophotometer was adjusted to the correct λmax and was blanked with water of the same pH as the test solution. HA solution with 0.01 g SA/100 mL H_2_O with pH 2.1 was used as a stock solution for creating a set of standard samples after serial dilutions with water of the same pH. Three scans were taken for each sample and the absorbance (AU) was plotted as a function of concentration (g/mL).

##### Determination of Drug Loading

Determination of SA loading was conducted with UV-Vis spectrophotometry in the saturated SA solutions pre- and post-loading. Following the 24 h loading in 10 mL aqueous saturated SA solutions, the films were removed from the vials and the solution’s absorbance was measured with a UV-Vis spectrophotometer (post-loading solution) and was compared with the pre-loading solution absorbance. The spectrophotometer was blanked with water at the same pH value and λmax as the before and after solutions. Three scans were taken for each sample with a resolution of 0.5 nm. According to Beer–Lambert law, the concentration and absorbance of a solution are directly proportional [12]. The ionization of HA and SA was also determined in order to support the elucidation of the drug loading results. The ionization (%) of HA in each pH was calculated from the Henderson–Hasselbalch equation [67].

#### 4.2.4. Fourier-Transform Infra-Red (FT-IR)

FTIR analysis was performed using a Shimadzu IR Affinity-1S Fourier Transform Infrared Spectrometer (Shimadzu UK Ltd., Milton Keynes, UK) on the following samples:Films swelled in water of different pH values (2.1, 7.8 and 11);Films loaded with SA solutions of different pH values (2.1, 7.8 and 11);SA powder.

A broad scan of the samples was performed from 550 cm^−1^ to 4000 cm^−1^ with a resolution of 2 wavenumbers (cm^−1^) in order to evaluate the potential loading of SA in the films (64 scans per sample). The spectra were recorded in transmittance mode at room temperature.

#### 4.2.5. Weight Studies

The drug loading procedure described in Section 4.2.1. supported complementary studies related to film swelling, gel fraction and film weight loss. More specifically, pre-weighted xerogels were loaded in water or saturated SA solutions of different pH values at 32 °C. After 24 h, specimens were removed from the solutions, gently blotted with filter paper to remove excess water and weighed. They were left to dry for 24 h in a vacuum oven (D-36-04-Vacucell, MMM Group, München, Germany) at 42 °C and were weighed again. The films were left in room environment conditions for 5 days and weight measurements were repeated. The drug loading (%) of SA within the films based on their weight was calculated according to the following equation:(3)$$\mathrm{Drug}\;\mathrm{loading}\;(\%)=\frac{\left(W_{\alpha }-W_{i}\right)}{W_{i}}\times 100$$
where W_i_ is the initial xerogel weight and W_α_ is the weight of the drug-loaded film [65].

Followingly, weight loss studies took place for the evaluation of the potential drug loading and film integrity in the solution. The film’s weight loss (%) was determined according to the following equation:(4)$$\mathrm{Weight}\;\mathrm{loss}\;(\%)=\frac{W_{i}-W_{\alpha }}{W_{i}}\times 100$$
where W_i_ is the initial xerogel weight and W_α_ is the film’s weight post-loading (dry state) [68].

The degree of swelling was calculated according to the following equation:(5)$$\mathrm{Swelling}\;(\%)=\frac{W_{s}-W_{i}}{W_{i}}\times 100$$
where W_i_ is the initial xerogel weight and W_s_ is the film’s weight post-loading (swollen film) [32].

The gel fraction for every film is calculated according to the following equation:(6)$$\mathrm{Gel}\;\mathrm{Fraction}\;(\%)=\frac{W_{w}}{W_{i}}\times 100$$
where W_i_ is the initial xerogel weight and W_w_ is the film’s weight post-loading (dry film) [32].

The relative difference between two values was calculated according to the following equation:(7)$$\mathrm{Relative}\;\mathrm{Change}\;(\%)=\frac{N_{2}-N_{1}}{N_{1}}\times 100$$
where N_1_ is the initial value of a measurement and N_2_ the value after a change occurred.

#### 4.2.6. Film Thickness

The films’ average thickness after swelling in water or loading in SA solutions was measured using a micrometer screw gauge (0–25 mm, 0.01 mm, Duratool, Taichung, Taiwan) at three randomly scattered points for each film. The results were expressed by the mean of 3 measurements ± standard deviation (SD). The increase (%) in film thickness was calculated according to Equation (10):(8)$$\mathrm{Thickness}\;\mathrm{increase}\;(\%)=\frac{\left(X_{1}-X_{2}\right)}{X_{2}}\times 100$$
where X_2_ is the initial thickness of the xerogel and X_1_ the film’s thickness after swelling or loading.

#### 4.2.7. Statistical Analysis

Data analysis and graphical representations were performed using EXCEL software (version 16.58). The average values and standard deviation calculated for n measurements were n ≥ 3. Differences between *t*-test groups were considered significant for *p* values < 0.05.

## Figures and Tables

**Figure 1 gels-10-00054-f001:**
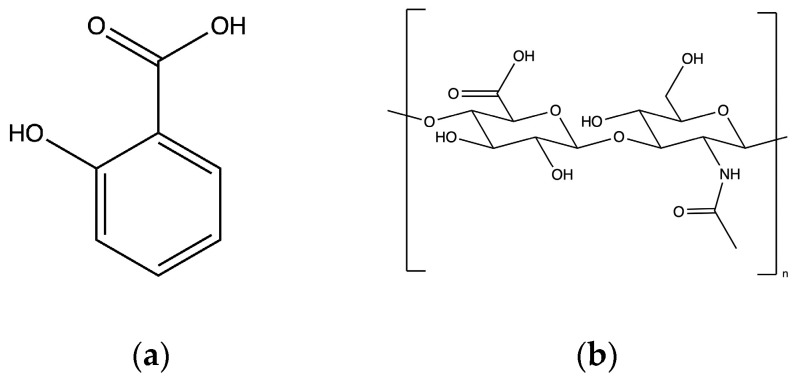
Chemical structures of (**a**) SA and (**b**) the disaccharide unit of HA.

**Figure 2 gels-10-00054-f002:**
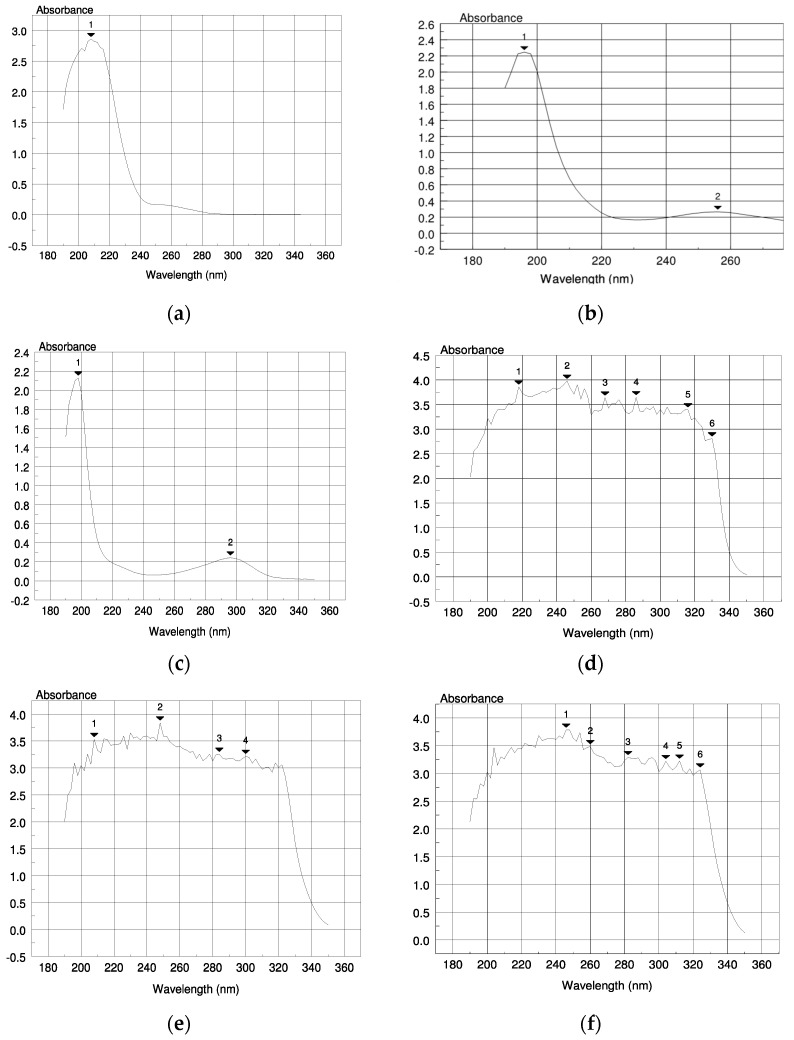
UV-Vis spectrum graphs for aqueous saturated solutions of (**a**) HA at pH 2.1, (**b**) HA at pH 7.8, (**c**) HA at pH 11, (**d**) SA at pH 2.1, (**e**) SA at pH 7.8, and (**f**) SA at pH 11.

**Figure 3 gels-10-00054-f003:**
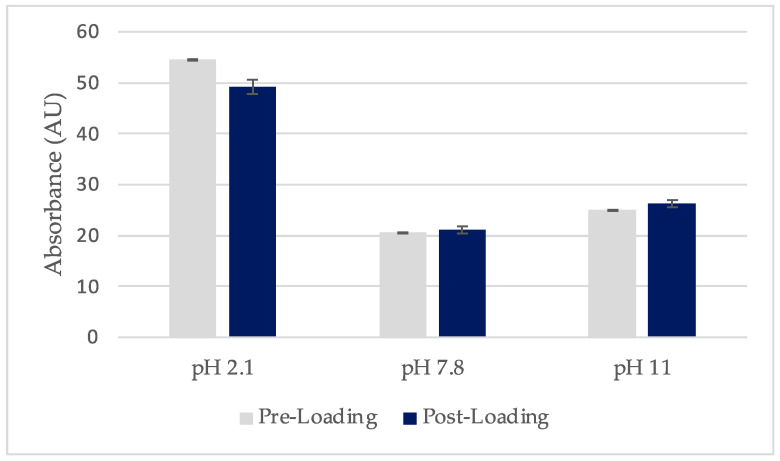
UV-Vis absorbance of saturated SA solutions pre- and post-loading of films at different pH values (±SD, n = 3–5).

**Figure 4 gels-10-00054-f004:**
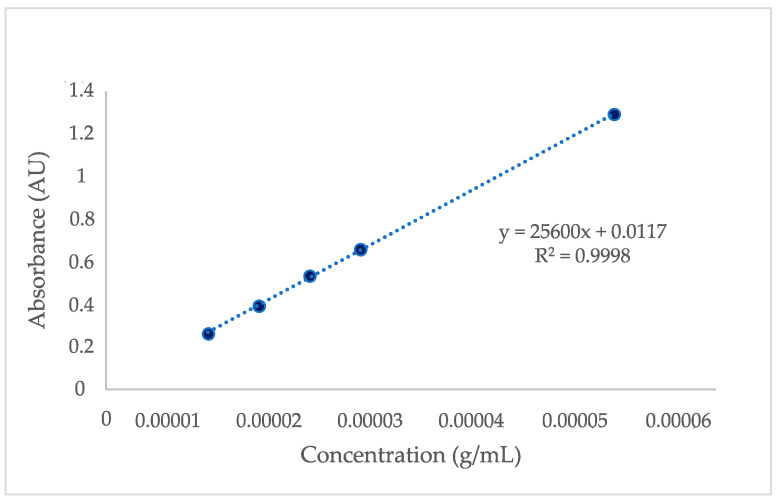
Calibration curve for SA at pH 2.1.

**Figure 5 gels-10-00054-f005:**
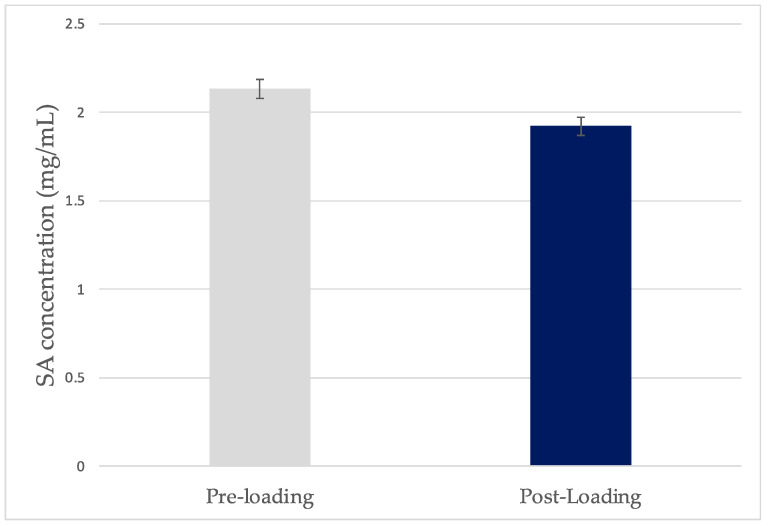
SA concentration in the pre- and post-loading solutions at pH 2.1 (±SD, n = 3).

**Figure 6 gels-10-00054-f006:**
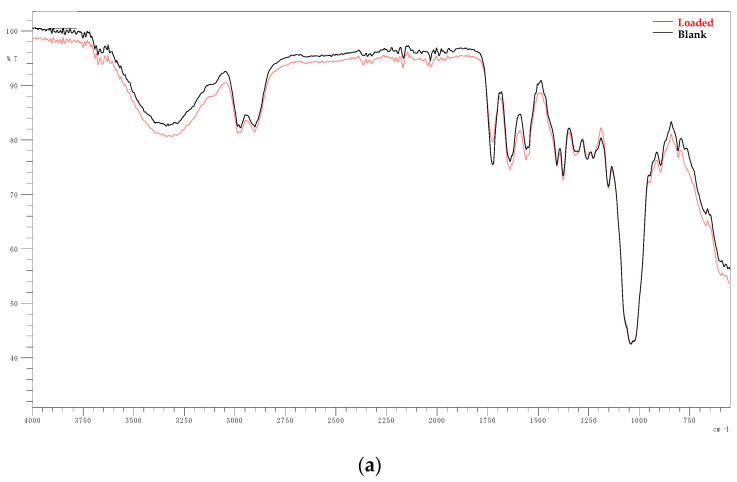
Comparison between the FT-IR spectra of films loaded in SA solutions and films loaded in water at pH: (**a**) 2.1, (**b**) 7.8 and (**c**) 11.

**Figure 7 gels-10-00054-f007:**
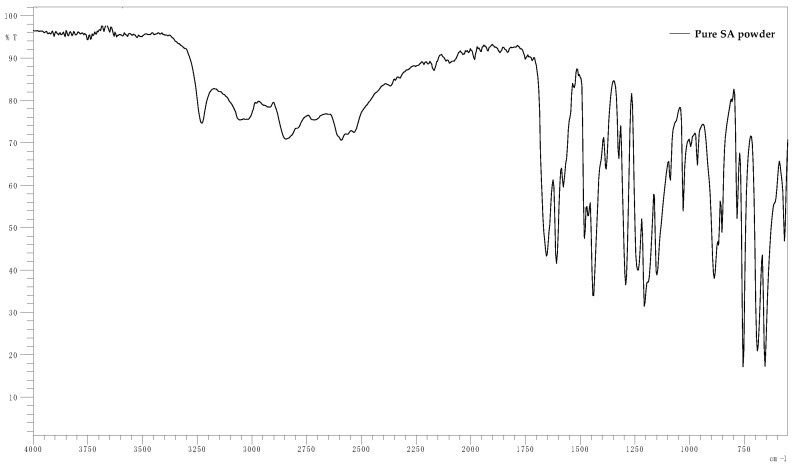
FT-IR spectra of pure SA powder.

**Figure 8 gels-10-00054-f008:**
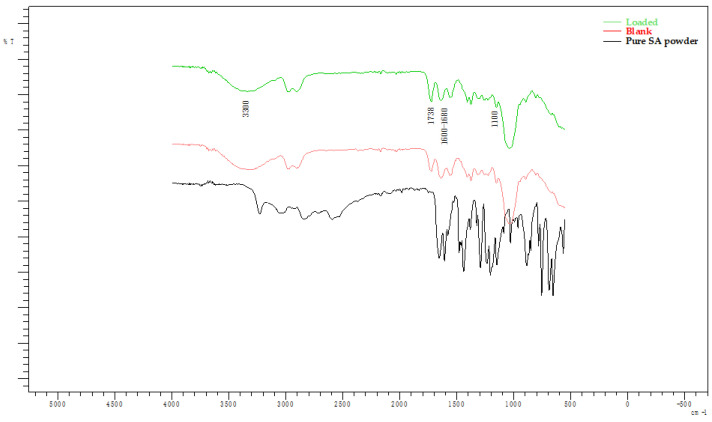
FT-IR spectra of samples loaded in SA solution (pH 2.1) and water (pH 2.1) compared to the spectra of pure SA powder.

**Figure 9 gels-10-00054-f009:**
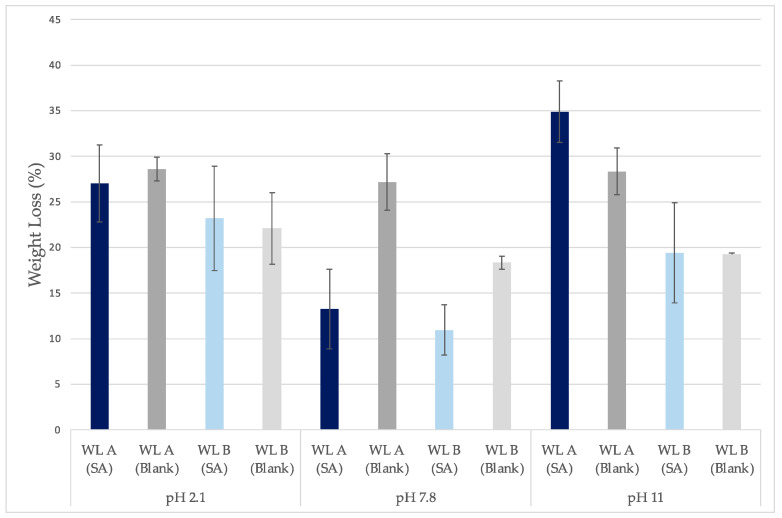
Comparison between weight loss A (WL_A_) and weight loss B (WL_B_) at each pH for films loaded in SA solutions (SA) or water (Blank) (±SD, n = 3–5).

**Figure 10 gels-10-00054-f010:**
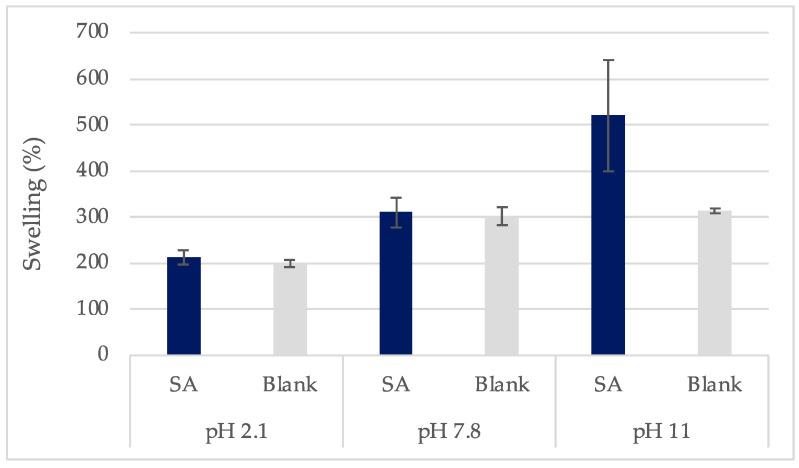
Comparison between films’ swelling (%) values at each pH for films loaded in SA solutions (SA) or water (Blank) (±SD, n = 3–5).

**Figure 11 gels-10-00054-f011:**
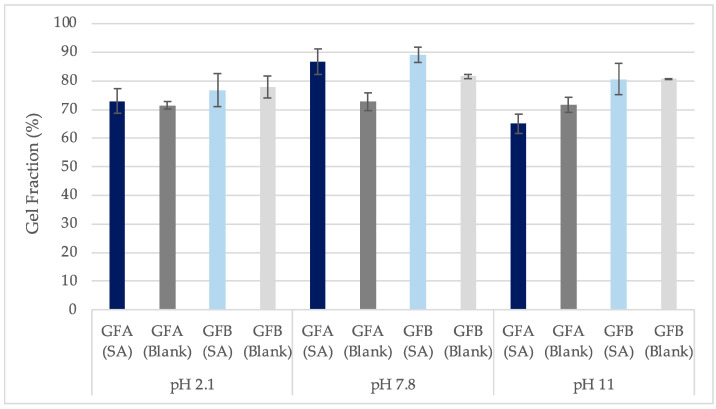
Comparison between gel fraction A (%) (GF_A_) and gel fraction B (%) (GF_B_) values at each pH for films loaded in SA solutions (SA) or water (Blank) (±SD, n = 3–5).

**Figure 12 gels-10-00054-f012:**
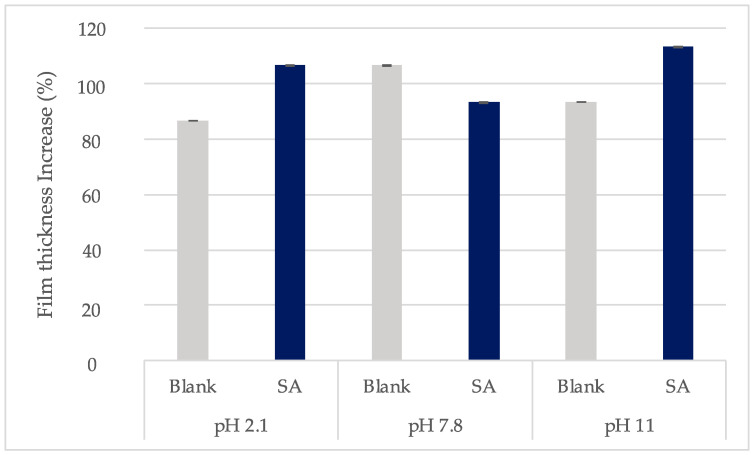
Thickness increase (%) of xerogels post-swelling or loading in solutions with different pH values (±SD, n = 3–5).

**Figure 13 gels-10-00054-f013:**
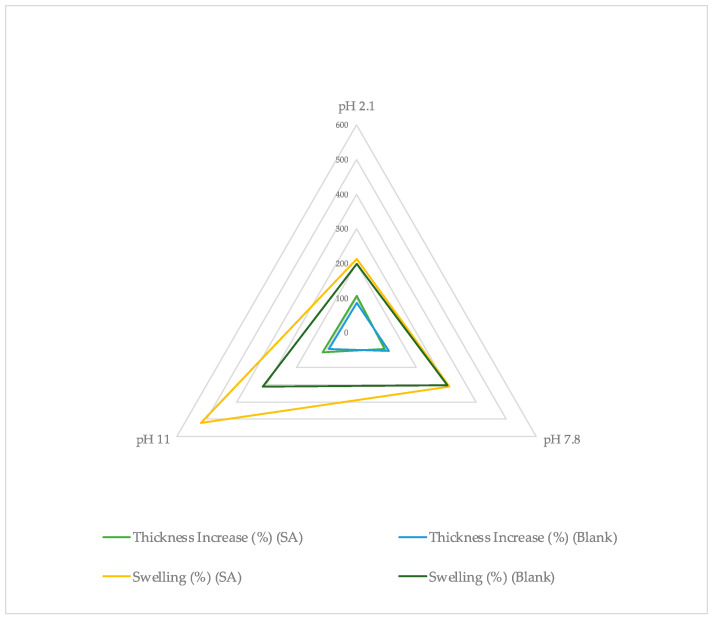
Average thickness and swelling increase (%) of xerogels post-swelling or loading in solutions with different pH values.

**Figure 14 gels-10-00054-f014:**
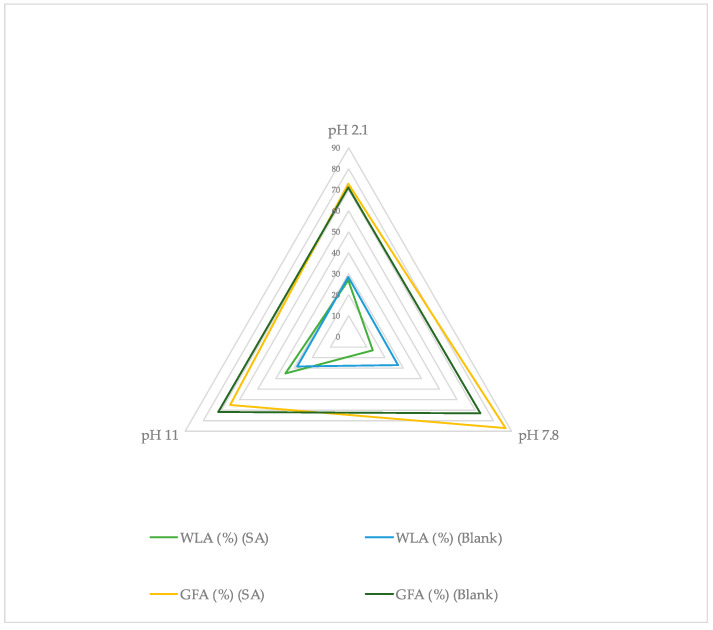
Average weight loss A and gel fraction A (%) of xerogels post-swelling or loading in solutions with different pH values.

**Table 1 gels-10-00054-t001:** Ionization degree and solubility of SA in water with different pH values at 37 °C [48].

pH	Solubility (mg/mL)	Ionized Salicylic Acid (%)	Temperature (°C)
2.00	15.3 ± 0.9	9.9	37
5.00	>34.9 ± 3.2	99.1	

**Table 2 gels-10-00054-t002:** Saturated solubility of SA in water at different pH values at 25 °C.

pH	Saturated Solubility (mg SA/mL H_2_O)
2.1	1.97
7.8	2.47
11	2.47

**Table 3 gels-10-00054-t003:** Ionization (%) of SA in solutions with different pH values.

pH	Ionization (%)
2.1	11.18
7.8	99.99
11	99.94

**Table 4 gels-10-00054-t004:** Weight measurements for the loaded or swelled films (n = 3–5).

Sample	pH	Average Film Weight (g) (±SD)			
		Xerogel Pre-Loading	Swelled Film Post-loading	Xerogel Post-Loading A	Xerogel Post-Loading B
Blank	2.1	0.0055 (±0.022)	0.0166 (±0.007)	0.0039 (±0.002)	0.0042 (±0.002)
Blank	7.8	0.0063 (±0.001)	0.0255 (±0.005)	0.0046 (±0.001)	0.0052 (±0.001)
Blank	11	0.0065 (±0.003)	0.0269 (±0.002)	0.0047 (±0.000)	0.0053 (±0.000)
SA	2.1	0.0060 (±0.001)	0.0187 (±0.003)	0.0044 (±0.001)	0.0046 (±0.001)
SA	7.8	0.0049 (±0.000)	0.0199 (±0.002)	0.0042 (±0.001)	0.0043 (±0.000)
SA	11	0.0050 (±0.001)	0.0317 (±0.010)	0.0033 (±0.000)	0.0040 (±0.000)

**Table 5 gels-10-00054-t005:** Weight loss (%), swelling (%) and gel fraction (%) as calculated for films (a) swelled in water or (b) loaded in SA solutions (n = 3–5).

Sample	pH	Degree of Swelling (%) (±SD)	Gel Fraction A_1_ (%) (±SD)	Gel Fraction B_2_ (%) (±SD)	Weight Loss A_1_ (%) (±SD)	Weight Loss B_2_ (%) (±SD)
Blank	2.1	198.70 (±7.79)	71.43 (±1.29)	77.92 (±3.90)	28.57 (±1.30)	22.08 (±3.39)
Blank	7.8	301.89 (±19.28)	72.81 (±3.11)	81.66 (±0.73)	27.19 (±3.12)	18.34 (±0.73)
Blank	11	312.86 (±4.79)	71.66 (±2.54)	80.76 (±0.12)	28.34 (±2.54)	19.24 (±0.12)
SA	2.1	212.63 (±16.29)	72.98 (±4.24)	76.80 (±5.72)	27.02 (±4.24)	23.20 (±5.73)
SA	7.8	310.32 (±32.41)	86.76 (±4.35)	89.06 (±2.76)	13.24 (±4.35)	10.94 (±2.76)
SA	11	520.12 (±121.64)	65.10 (±3.37)	80.60 (±5.49)	34.90 (±3.37)	19.40 (±5.49)

1. Weight loss A_1_ and gel fraction A_1_ are calculated based on the initial xerogel weight and its dry weight after loading and drying in the oven. 2. Weight loss B_2_ and gel fraction B_2_ are calculated based on the initial xerogel weight and its dry weight after loading, drying in the oven and conditioning in room environment conditions for 5 days.

**Table 6 gels-10-00054-t006:** Relative difference (%) between the average values of weight loss A (WL_A_) and weight loss B (WL_B_) and gel fraction A (GF_A_) and gel fraction B (GF_B_) (n = 3–5).

Sample	pH	Difference WL_A_-WL_B_ (%) (±SD)	Difference GF_A_-GF_B_ (%) (±SD)
Blank	2.1	−23.03 (±10.15)	9.05 (±3.47)
Blank	7.8	−31.76 (±8.75)	12.33 (±5.33)
Blank	11	−31.74 (±6.56)	12.80 (±4.14)
SA	2.1	−19.89 (±8.36)	7.13 (±2.20)
SA	7.8	−28.75 (±12.37)	5.28 (±4.32)
SA	11	−44.62 (±13.33)	23.85 (±6.75)

**Table 7 gels-10-00054-t007:** Thickness of xerogels (before swelling) and hydrogels (after loading/swelling).

	Sample	pH	Film Thickness (mm) (n = 3)	Increase of Film Thickness (%)
Xerogel			0.150	N/A
Swollen unmedicated film (Blank)	Blank 1	2.1	0.213	86.667
	Blank 2	7.8	0.320	106.667
	Blank 3	11	0.293	93.333
Loaded Film (SA)	SA 1	2.1	0.310	106.667
	SA 2	7.8	0.290	93.333
	SA 3	11	0.323	113.333

**Table 8 gels-10-00054-t008:** SA solutions of different pH values and blanking solution used.

Solution	pH	Blanking Solution
SA Solution 1	2.1	Water at pH 2.1
SA Solution 2	7.8	Water at pH 7.8
SA Solution 3	11	Water at pH 11

## Data Availability

Data are contained within the article.

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
