# Peer review of "Studies on Loading Salicylic Acid in Xerogel Films of Crosslinked Hyaluronic Acid"

_gels, 2024, doi:10.3390/gels10010054_

Round 1

Reviewer 1 Report

Comments and Suggestions for Authors

Current work on salicylic acid loading in xerogel.  The article sufficiently contains novelty. 

Accept with the following correction;

1. Table 3 and Figure 2 contain the same information then remove it and provide supplementary data.

2. suggestion, if possible then make the calibration curve of a drug in respective media, and based on the linearity equation concentration is ideal. 

3. Table 4 and Table 3 pass the same information, kindly remove one either table or image. here give the drug content in place of absorbance 

4. Figure 4, line title needs to be rewritten as per the result. 

5. Data showing the SD bar but figures and table legends not mention.  The result must be present means plus minus sd bar, n=......

Author Response

Accept with the following correction:

  1. Table 3 and Figure 2 contain the same information then remove it and provide supplementary data.

Thank you for the comment. Table 3 was removed and the text was modified accordingly. 

2. suggestion, if possible then make the calibration curve of a drug in respective media, and based on the linearity equation concentration is ideal. 

Thank you for the comment. We prepared a calibration curve for SA at pH 2.1, as according to the UV-Vis studies it was the most promising pH for drug-loading.  The method was also added in paragraph 4.2.3.2.

3. Table 4 and Table 3 pass the same information, kindly remove one either table or image. here give the drug content in place of absorbance 

Thank you for the comment. We believe that the reviewer refers to Table 4 and Figure 3; therefore Table 4 was removed and Figure 3 was left on the manuscript.

The drug content (concentration and mass fraction of drug in the xerogel) was calculated for pH 2.1 where the UV-Vis study showed the most promising results.

4. Figure 4, line title needs to be rewritten as per the result. 

Thank you for the comment. The line titles for the FT-IR graphs have been modified.

5. Data showing the SD bar but figures and table legends not mention.  The result must be present means plus minus sd bar, n=......

Thank you for the observation. The tables and figures have been updated based on the above comment.

Reviewer 2 Report

Comments and Suggestions for Authors

The article by Mamaligka A.M. and Dodou K. describes a study of efforts to load cross-linked sodium hyaluronate films with salicylic acid. The authors have tried loading salicylic acid by swelling a film of cross-linked sodium hyaluronate in an aqueous salicylic acid solution at three different pH values. Then the authors use UV-vis spectrophotometry, IR spectroscopy, and weight-based methods to evaluate the best approach for loading salicylic acid. As a result, the authors show that the most promising outcome is obtained at a low pH value, although they do not characterize it numerically. The article is clearly written and contains practical findings that may be of interest to those involved in cosmetic chemistry. Nevertheless, the article requires revisions before publication.

Specific comments are as follows.

Line 38: “the gel maintains its porous structure”. This phrase appears odd. The authors write about gels obtained by cross-linking polymers in their solution (e.g., hyaluronic acid). However, polymer solutions do not have pores even after crosslinking. Gels can have pores if they are formed from colloid particles, such as nanocellulose gels, silica gels, clay gels, and so on. However, the authors write about polymer gels rather than colloid ones. In other words, "its porous structure" is better replaced by "its network structure", meaning a network of cross-links. The same is true on line 42: “hydrogel’s pores is evaporating” is better replaced by “hydrogel’s network is evaporating”.

Line 77: “cosmetic formulations and dermal filler injections”. A reference to the literature needs to be added, e.g., 10.1007/s00397-016-0913-z.

Line 119: “SA has high aqueous solubility”. Do the authors seriously believe that the solubility listed in Table 1 is high? In the reviewer's opinion, it is low solubility. The same is true on line 128: “high aqueous solubility”.

Table 1. The temperature needs to be specified. In addition, please check the solubility values, as they seem to be overestimated by a factor of 10.

Line 142: “2.47 g/mL”. The solubility cannot be so high in this case. It probably is 2.47 mg/mL. And this quantity is 10 times lower than that presented in Table 1.

Line 146: “at pH values higher than 2.97 SA loses H+ from its hydroxy- and carboxy- groups”. At this pH, only the carboxyl group dissociates. The hydroxyl group has its own pKa, which is much higher than 2.97 (about 13.8).

Line 298: “between the carbonyl group”. The authors are probably referring to the carboxyl group.

Line 460: “At pH 2.1 the drug loading was promising”. The mass fraction of salicylic acid in the final xerogel is lacking. Could the authors provide an estimate?

Line 475: “HA sodium salt”. The authors use hyaluronic acid salt. However, the text of the article discusses hyaluronic acid rather than its salt. For example, line 173: "HA losed more protonated ions (H+)". This contradiction should be eliminated.

Line 480: “crosslinked hyaluronic acid xerogels were prepared according to the method described”. Nevertheless, a brief description should be given: crosslinking agent, its concentration, crosslinking conditions, gel after-crosslinking purification procedures, gel drying conditions, and so on.

Line 553: “The ionization (%) of HA in each pH was calculated from the Henderson-Hasselbalch equation for weak acids:” The authors use sodium hyaluronate but write the formula for hyaluronic acid. This contradiction should be resolved.

Comments on the Quality of English Language

The English language requires moderate editing.

Author Response

Specific comments are as follows.

Line 38: “the gel maintains its porous structure”. This phrase appears odd. The authors write about gels obtained by cross-linking polymers in their solution (e.g., hyaluronic acid). However, polymer solutions do not have pores even after crosslinking. Gels can have pores if they are formed from colloid particles, such as nanocellulose gels, silica gels, clay gels, and so on. However, the authors write about polymer gels rather than colloid ones. In other words, "its porous structure" is better replaced by "its network structure", meaning a network of cross-links. The same is true on line 42: “hydrogel’s pores is evaporating” is better replaced by “hydrogel’s network is evaporating”.

Thank you for the insightful comment. This has now been corrected on the revised manuscript.

Line 77: “cosmetic formulations and dermal filler injections”. A reference to the literature needs to be added, e.g., 10.1007/s00397-016-0913-z.

Thank you for the comment. The relevant references both for cosmetic formulations and dermal filler injections have now been added.

Line 119: “SA has high aqueous solubility”. Do the authors seriously believe that the solubility listed in Table 1 is high? In the reviewer's opinion, it is low solubility. The same is true on line 128: “high aqueous solubility”.

Thank you for your comment. This has now been corrected in the text. Indeed, the aqueous solubility of SA is not high, but it is pH dependent.

Table 1. The temperature needs to be specified. In addition, please check the solubility values, as they seem to be overestimated by a factor of 10.

Thank you for your comment, indeed it's important to state the temperature considering that the solubility is temperature-dependent. The temperature has now been listed in the text and additionally added on table 1. The solubility values in table 1 were taken from the literature (Table 1 from the following reference: Otto, D.P.; Combrinck, J.; Otto, A.; Tiedt, L.R.; de Villiers, M.M., Dissipative Particle Dynamics Investigation of the Transport of Salicylic Acid through a Simulated In Vitro Skin Permeation Model. Pharmaceuticals, 2018, 11(4), 134, DOI: 10.3390/ph11040134).

Line 142: “2.47 g/mL”. The solubility cannot be so high in this case. It probably is 2.47 mg/mL. And this quantity is 10 times lower than that presented in Table 1.

Thank you for your comment. ‘2.47 g/mL’ has been corrected to ‘2.47 mg/mL’. The solubility values presented in table 1 were determined at 37 oC, thus, they are expected to be higher compared to the values we determined at 25 oC due to the increase of solubility in higher temperature.

Line 146: “at pH values higher than 2.97 SA loses H+ from its hydroxy- and carboxy- groups”. At this pH, only the carboxyl group dissociates. The hydroxyl group has its own pKa, which is much higher than 2.97 (about 13.8).

Thank you for the insightful comment. The ‘Hydroxyl group’ has been removed from the text.

Line 298: “between the carbonyl group”. The authors are probably referring to the carboxyl group.

Thank you, this was a typo and it has now been corrected.

Line 460: “At pH 2.1 the drug loading was promising”. The mass fraction of salicylic acid in the final xerogel is lacking. Could the authors provide an estimate?

Thank you for the comment. For the calculation of the mass fraction the calibration of SA at pH 2.1 was plotted and the concentrations of the pre- and post-loading solutions were calculated from the linear regression equation. The mass fraction of SA in the xerogel can be found in paragraph 2.2.2.

Line 475: “HA sodium salt”. The authors use hyaluronic acid salt. However, the text of the article discusses hyaluronic acid rather than its salt. For example, line 173: "HA losed more protonated ions (H+)". This contradiction should be eliminated.

Thank you, we have now corrected this inaccuracy from our article. The sodium carboxylate group is not reacting during the cross linking reaction, therefore it remains on the crosslinked network structure and it is susceptible to protonation at low pH values. 

Line 480: “crosslinked hyaluronic acid xerogels were prepared according to the method described”. Nevertheless, a brief description should be given: crosslinking agent, its concentration, crosslinking conditions, gel after-crosslinking purification procedures, gel drying conditions, and so on.

Thank you, we have now added these details.

Line 553: “The ionization (%) of HA in each pH was calculated from the Henderson-Hasselbalch equation for weak acids:” The authors use sodium hyaluronate but write the formula for hyaluronic acid. This contradiction should be resolved.

Thank you, we have now addressed this in our revised manuscript.

Reviewer 3 Report

Comments and Suggestions for Authors

General Comments

The submitted manuscript is focused on the investigation of the salicylic acid loading potential in hyaluronic acid based crosslinked hydrogel films using a post-loading (osmosis) method of the unmedicated xerogels from saturated aqueous solutions of salicylic acid over a range of pH values.

The topic of this work is original and worthy of investigation. It well matches the Gels aim and scope, but some revisions have to be applied, as specified below.

Moreover, an accurate English language revision is strongly mandatory

Specific suggestions are reported below point by point.

Abstract

All the acronyms, such as FTIR, have to be specified the first Tim they are used.

INTRODUCTION

-        The following sentences “The latest findings enabled the development of novel patches and the incorporation of numerous ingredients. However, more studies should be conducted in order to unlock the limitless potentials in this field.” have to be supported with suitable literature references, including “Hydrogen sulfide-releasing fibrous membranes: Potential patches for stimulating human stem cells proliferation and viability under oxidative stress. International Journal of Molecular Sciences, 19(8)(2018), 2368.”.

-        The statement “Hydrogels are an effective and convenient vehicle for the loading and release of active ingredients.” needs suitable references, including “Biosynthesis of innovative calcium phosphate/hydrogel composites: Physicochemical and biological characterisation. Nanotechnology, 32(9) (2020), 095102” and “Injectable silk fibroin hydrogels functionalized with microspheres as adult stem cells-carrier systems. International journal of biological macromolecules, 108(2018), 960-971.”.

-        It is strongly suggested to add a brief list of the used characterisations, at the end of the Introduction section.

2. Results and Discussion

2.2. UV-Vis spectrophotometry

2.2.1. Calculation of λmax

-        - The following considerations “As the pH value of the solu-172 tion increased, HA losed more protonated ions (H+) which were released in the solution, 173 absorbed light and raised the maximum absorbance of the solution. This phenomenon 174 was more intense in the case of HA compared to SA as it is a poly-anion and has more 175 groups that can lose protons (hydroxy-, carboxy- groups) (Figure 1)” have to be supported with proper literature references.

- The consideration “This could be due to instrumental error as the concentration and absorbance post-loading is expected to be the same or less compared to pre-loading.” needs proper references.

- The statement “The lower ionization of the compounds in pH<pKa could have permitted the drug loading of SA” has to be supported with suitable references.

2.3. Fourier-Transform Infra-Red (FT-IR)

- All the reported spectra should be better and more deeply described, assigning all the peaks. The peaks assignments have to be supported with proper references.

- In the related Figures, the main peask should be assigned.

2.4. Weight studies

- The consideration “As a result, calculation of Drug Loading (%) from equation 6 generated negative values. This is indicative of the disintegration that took place” needs suitable references.

- The statement “The noticed weight loss was attributed to the partial disintegration of the films in the solutions due to multiple factors” has to be corroborated with proper references.

- The sentences “These data showcase the importance of storing hyaluronic acid films in sealed containers to avoid excessive water uptake during experiments or throughout their commercial shelf life. Higher WLA-WLB difference indicates higher moisture uptake from the environment and could be related to more significant alterations in the films’ 3D-network that enabled more units of HA to bind with water (Table 8).” have to be supported wit suitable references.

- The following considerations “This, was in accordance to the higher weight loss for the films and indicated a potential destructive effect of SA on the HA-chains and the bonds in the 3D-network of the cross-linked films. The pH-dependent swelling values of the films were due to the different ionic strength of each loading solution. HA is a weak polyelectrolyte and, thus, is highly sensitive to pH and ionic strength. HA has a pka of 3.0 and at pH 11 344 it loses H+ and is highly ionized compared to neutral or acidic pH values where it is less on non-ionized, respectively (Table 5)” need suitable references.

- The consideration” This was indicative of lower disintegration compared to the highly basic and acidic solutions” has to be supported with proper references.

4. Materials and Methods

4.1. Materials

Even if reported elsewhere, the preparation procedure for the crosslinked hyaluronic acid xerogels has to be briefly reported.

4.2. Methods

4.2.1. Drug Loading Method

- The dimensions of the pre-cut rectangular xerogels samples have to be reported.

- The number of the performed experiments and analysed samples in order to provide and average value has to be specified.

4.2.2. Determination of SA Saturated Solubility

4.2.2.1. Preparation of acidic SA saturated solution.

- The reported equations have to be supported with appropriate references.

4.2.3.4. Determination of Drug Loading

-        The resolution has to be specified.

-        The details about the Lambert Beer law and the Henderson-Hasselbalch equation could be removed, since well known and too didactical.

-        - On the other hand, the evaluated SA concentrations in order to obtain the calibration curve have to be specified.

4.2.4. Fourier-Transform Infra-Red (FT-IR)

. The number of scans has to be specified.

Comments on the Quality of English Language

The English language requires moderate revisions.

Author Response

The topic of this work is original and worthy of investigation. It well matches the Gels aim and scope, but some revisions have to be applied, as specified below.

Moreover, an accurate English language revision is strongly mandatory

Specific suggestions are reported below point by point.

Abstract

All the acronyms, such as FTIR, have to be specified the first Tim they are used.

 Thank you, the acronyms have now been specified.

INTRODUCTION

  •        The following sentences “The latest findings enabled the development of novel patches and the incorporation of numerous ingredients. However, more studies should be conducted in order to unlock the limitless potentials in this field.” have to be supported with suitable literature references, including “Hydrogen sulfide-releasing fibrous membranes: Potential patches for stimulating human stem cells proliferation and viability under oxidative stress. International Journal of Molecular Sciences, 19(8)(2018), 2368.”.

These references have now been added.

  •        The statement “Hydrogels are an effective and convenient vehicle for the loading and release of active ingredients.” needs suitable references, including “Biosynthesis of innovative calcium phosphate/hydrogel composites: Physicochemical and biological characterisation. Nanotechnology, 32(9) (2020), 095102” and “Injectable silk fibroin hydrogels functionalized with microspheres as adult stem cells-carrier systems. International journal of biological macromolecules, 108(2018), 960-971.”.

These references have now been added.

  •        It is strongly suggested to add a brief list of the used characterisations, at the end of the Introduction section.

Thank you, it's been added.

  1. Results and Discussion

2.2. UV-Vis spectrophotometry

2.2.1. Calculation of λmax

  •        - The following considerations “As the pH value of the solu-172 tion increased, HA losed more protonated ions (H+) which were released in the solution, 173 absorbed light and raised the maximum absorbance of the solution. This phenomenon 174 was more intense in the case of HA compared to SA as it is a poly-anion and has more 175 groups that can lose protons (hydroxy-, carboxy- groups) (Figure 1)” have to be supported with proper literature references.

This has now been corrected and it is explained clearly in the revised manuscript (lines 184-191).

  • The consideration “This could be due to instrumental error as the concentration and absorbance post-loading is expected to be the same or less compared to pre-loading.” needs proper references.

 The salicylic acid concentration in the aqueous solution post-loading can’t be higher than the concentration pre-loading; the concentration post-loading can be either lower (if loading occurred) or the same (no loading). 

  • The statement “The lower ionization of the compounds in pH<pKa could have permitted the drug loading of SA” has to be supported with suitable references.

Thank you for your comment. The relevant reference has now been added.

2.3. Fourier-Transform Infra-Red (FT-IR)

  • All the reported spectra should be better and more deeply described, assigning all the peaks. The peaks assignments have to be supported with proper references.

Thank you for your comment. The main peaks have been added to figure 8 and additional details with the supporting references have been added in the main text.

  • In the related Figures, the main peask should be assigned.

Please see revised Figures 6, 7, 8.

2.4. Weight studies

  • The consideration “As a result, calculation of Drug Loading (%) from equation 6 generated negative values. This is indicative of the disintegration that took place” needs suitable references.

These paragraphs have now been revised, please see revised manuscript.

  • The statement “The noticed weight loss was attributed to the partial disintegration of the films in the solutions due to multiple factors” has to be corroborated with proper references.

In-depth referenced explanations have now been added.

  • The sentences “These data showcase the importance of storing hyaluronic acid films in sealed containers to avoid excessive water uptake during experiments or throughout their commercial shelf life. Higher WLA-WLB difference indicates higher moisture uptake from the environment and could be related to more significant alterations in the films’ 3D-network that enabled more units of HA to bind with water (Table 8).” have to be supported wit suitable references.

Thank you. Please see revised explanations on the manuscript.

  • The following considerations “This, was in accordance to the higher weight loss for the films and indicated a potential destructive effect of SA on the HA-chains and the bonds in the 3D-network of the cross-linked films. The pH-dependent swelling values of the films were due to the different ionic strength of each loading solution. HA is a weak polyelectrolyte and, thus, is highly sensitive to pH and ionic strength. HA has a pka of 3.0 and at pH 11 344 it loses H+ and is highly ionized compared to neutral or acidic pH values where it is less on non-ionized, respectively (Table 5)” need suitable references.

Corrections have been made on the above statements, please see revised manuscript.

  • The consideration” This was indicative of lower disintegration compared to the highly basic and acidic solutions” has to be supported with proper references.

This statements has now been further explained in the revised manuscript. 

  1. Materials and Methods

4.1. Materials

Even if reported elsewhere, the preparation procedure for the crosslinked hyaluronic acid xerogels has to be briefly reported.

These details have now been added.

4.2. Methods

4.2.1. Drug Loading Method

  • The dimensions of the pre-cut rectangular xerogels samples have to be reported.

The dimensions have now been added.

  • The number of the performed experiments and analysed samples in order to provide and average value has to be specified.

This clarification has now been added.

4.2.2. Determination of SA Saturated Solubility

4.2.2.1. Preparation of acidic SA saturated solution.

  • The reported equations have to be supported with appropriate references.

We deduced these simple equations for the purpose of our studies, based on our experimental procedure.

4.2.3.4. Determination of Drug Loading

  •        The resolution has to be specified.

Thank you, the resolution of the spectrophotomer has now been added.

  •        The details about the Lambert Beer law and the Henderson-Hasselbalch equation could be removed, since well known and too didactical.

Thank you, these equations have now been removed.

  •        - On the other hand, the evaluated SA concentrations in order to obtain the calibration curve have to be specified.

Thank you, we have added this information in our revised manuscript.

4.2.4. Fourier-Transform Infra-Red (FT-IR)

. The number of scans has to be specified.

We have now specified the number of scans.

Round 2

Reviewer 2 Report

Comments and Suggestions for Authors

The authors have improved the article for its publication.

Comments on the Quality of English Language

The English language requires moderate editing.

Reviewer 3 Report

Comments and Suggestions for Authors

The revised manuscript looks very improved and can be accepted in the current version.

Comments on the Quality of English Language

The English is good